# Antibacterial Activity of Moroccan Zantaz Honey and the Influence of Its Physicochemical Parameters Using Chemometric Tools

**Youssef Elamine** [1,2,*], **Hamada Imtara** [3], **Maria Graça Miguel** [4], **Ofélia Anjos** [5,6,7], **Letícia M. Estevinho** [8], **Manuel Alaiz** [1], **Julio Girón-Calle** [1], **Javier Vioque** [1], **Jesús Martín** [9] and **Badiâa Lyoussi** [2]

1   Instituto de la Grasa (C.S.I.C.), Universidad Pablo de Olavide, Edificio 46, Carretera de Utrera, km 1, 41013 Sevilla, Spain; alaiz@ig.csic.es (M.A.); jgiron@ig.csic.es (J.G.-C.); jvioque@ig.csic.es (J.V.)
2   Laboratory of Natural Substances, Pharmacology, Environment, Modeling, Health and Quality of Life (SNAMOPEQ), University of Sidi Mohamed Ben Abdellah, Fez 30000, Morocco; badiaa.lyoussi@usmba.ac.ma
3   Faculty of Arts and Sciences, Arab American University Palestine, Jenin P.O. Box 240, Palestine; Hamada.imtara@aaup.edu
4   Department of Chemistry and Pharmacy, Mediterranean Institute for Agriculture, Environment and Development, Faculty of Science and Technology, Campus de Gambelas, University of Algarve, 8005-139 Faro, Portugal; mgmiguel@ualg.pt
5   Instituto Politécnico de Castelo Branco, 6001-909 Castelo Branco, Portugal; ofelia@ipcb.pt
6   Centro de Estudos Florestais, Instituto Superior de Agronomia, Universidade de Lisboa, Tapada da Ajuda, 1349-017 Lisboa, Portugal
7   Centro de Biotecnologia de Plantas da Beira Interior, 6001-909 Castelo Branco, Portugal
8   Centro de Investigação de Montanha (CIMO), Instituto Politécnico de Bragança, 5300-252 Bragança, Portugal; leticia@ipb.pt
9   Fundación MEDINA, Avda del Conocimiento 34, 18016 Granada, Spain; jesus.martin@medinaandalucia.es
*   Correspondence: youssef.elamine@usmba.ac.ma

**Abstract:** The emergence of multidrug-resistant bacteria has prompted the development of alternative therapies, including the use of natural products with antibacterial properties. The antibacterial properties of Zantaz honey produced in the Moroccan Atlas Mountains against *Escherichia coli*, *Pseudomonas aeruginosa*, and *Staphylococcus aureus* was evaluated and analyzed using chemometric tools. Minimum inhibitory concentration (MIC) and Minimum bactericidal concentration (MBC) against *S. aureus* were the lowest (112.5 ± 54.5 mg/mL), revealing that this species was most sensitive to Zantaz honey. *P. aeruginosa* showed an intermediate sensitivity (MIC= 118.75 ± 51.9 mg/mL), while *E. coli* was the most resistant to treatment (MIC = 175 ± 61.2 mg/mL). Content of monosaccharides, certain minerals, and phenolic compounds correlated with antibacterial activity ($p < 0.05$). Principal component analysis of physicochemical characteristics and antibacterial activity indicated that the parameters most associated with antibacterial activity were color, acidity, and content of melanoidins, fructose, epicatechin, methyl syringate, 4-coumaric acid, and 3-coumaric acid.

**Keywords:** *Bupleurum spinosum*; methyl syringate; *Escherichia coli*; *Pseudomonas aeruginosa*; *Staphylococcus aureus*

## 1. Introduction

Honey is one of the oldest foods used in traditional medicine for the treatment of different human ailments, including infectious diseases [1,2]. The antibacterial activity of honey was reported by Van Ketel in 1892 [3], and many studies have described antimicrobial activity in honeys with different botanical origins and the underlying mechanisms [4–6]. Nowadays, the emergence of multi-drug-resistant bacteria has promoted the reintroduction of natural products, including honey, as antibacterial agents [7].

The mechanism for the antibacterial activity of honey is not yet fully understood. It has been reported that degradation of bacterial DNA is promoted by the presence in

honey of hydrogen peroxide, phenolics, flavonoids, and other unknown components that act as pro-oxidants [8,9]. The sensitivity of bacteria to honey appears to be related to certain physicochemical characteristics [10] that include sugars, protein, melanoidin, and minerals, as well as water activity, acidity, viscosity, color, and conductivity [11,12]. For example, it has been reported that the antibacterial effect of honey increases with increasing content of water and reducing sugars, while it decreases with increasing electrical conductivity (EC) [13]. Further, a high correlation between conductivity and mineral content with antibacterial activity that is boosted by the presence of essential oils has been described [12]. Methylglyoxal and bee defensin-1 have also been reported to be involved in the antimicrobial activity of some honeys [14]. In addition, the parameters that determine the antibacterial activity of honey depend on the specific bacterial strain that is targeted. For example, minimum pH values of 4.3, 4.0, and 4.4 are required for inhibiting the growth of the pathogenic bacteria *Salmonella* spp., *Pseudomonas aeruginosa*, and *Streptococcus pyogenes*, respectively [5].

Zantaz honey is a dark-colored, monofloral honey from the prickly shrub *Bupleurum spinosum* Gouan (Apiaceae) that is produced in the Atlas Moroccan Mountains and is considered a traditional geographical specialty by the authorities. The scientific characterization of its health-promoting properties is of great interest in order to promote its consumption, which would help the economy of local low-income families. The cellular antioxidant and antiproliferative activities of this honey have been recently described [15]. The objective of this work was to determine the possible antibacterial activity of Zantaz honey, and to explore whether it correlates with physicochemical parameters.

## 2. Materials and Methods

### 2.1. Chemicals

Chemical standards used in the evaluation of polyphenolic compounds were purchased from Sigma, with an analytical purity grade higher than 98%. Ultrapure water was produced from a Milli-Q system (Millipore, Bedford, MA, USA), and methanol, acetonitrile, and 2-propanol were purchased from Teknokroma (Barcelona, Spain). All other chemicals were of analytical grade.

### 2.2. Zantaz Honey Sampling and Mellissopalynology

Ten honey samples were purchased from local beekeepers in the region of Fez-Meknes and were stored in the dark at room temperature. Namely, the samples were obtained from Ait Bazza (ZH1 and ZH2), Imouzzer Marmoucha (ZH3), Oulad Ali (ZH4 and ZH5), Jbel Bouiblane (ZH6), Timahdite (ZH9) and Ait Bouilloul (ZH7, ZH8 and ZH10) (Figure 1). The region are the high summits of the middle Atlas mountains in a circle of 150 Km diameter. Analysis of pollen species was carried out according to recommendations from the International Commission for Bee Botany (ICBB) as described previously [16]. An optic microscope (Leitz Messtechnik GmbH, Wetzlar, Germany) equipped with 400× and 1000× objectives was used for pollen identification and count. One thousand pollen grains were counted, and frequent classes were counted twice. Only dominant (more than 45%), secondary (between 16% and 45%), and important minor pollens (from 3 to 15%) were considered for the present study.

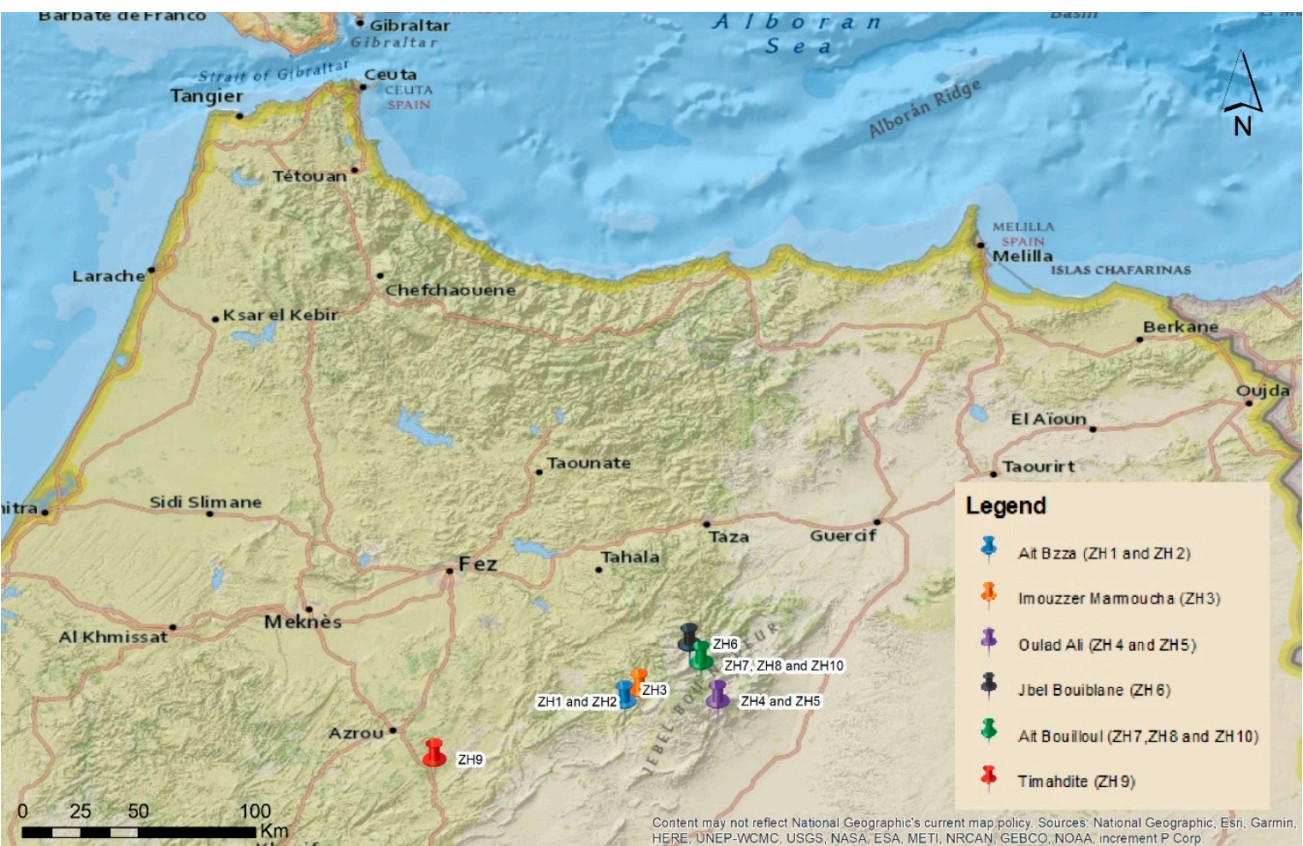

**Figure 1.** Study area map including the geographic sampling distribution.

### 2.3. Zantaz Honey Physicochemical Characterization

Acidity (free, lactonic and total), pH, ash content, electrical conductivity, water content, and diastase activity were determined as recommended by the Harmonized Methods from the International Honey Commission [17]. Color intensity was estimated on the Pfund scale by determination of absorbance of a 50% honey solution in water ($v/v$) at 635 nm using a Shimadzu spectrophotometer [18]. The mm Pfund value and honey color were calculated using the following algorithm: mm Pfund = $-38.7 + 371.39 \times Abs_{635}$ nm. Additionally, the color of honey was determined spectrophotometrically by calculating net absorbance ($Abs_{560}$ nm–$Abs_{720}$ nm). The estimation of melanoidin content was based on the browning index ($Abs_{450}$ nm–$Abs_{720}$ nm) [19].

Sugars were analyzed by High Performance Liquid Chromatography (HPLC) coupled with a refraction index detector according to Bogdanov (2009) [17]. An analytical stainless-steel column Purospher® STAR–NH2, 4 mm diameter, 250 mm length, and 5 μm particle size was used. The mobile phase was acetonitrile: water (80/20, $v/v$) at a flow rate of 1.3 mL/min. The column and detector were kept at 30 °C and injection volume was 10 μL. Sugars were identified and quantified by comparison with retention times and peak areas of sugar standards. Results are expressed as g/100 g honey.

Samples (5 g) were incinerated at 550 °C for determination of mineral content. After cooling, 5 mL 0.1 M nitric acid was added to the ashes and the mixture was taken to almost complete dryness by stirring on a heating plate. Then, 10 mL of the same nitric acid solution was added, and the mixture was brought up to a final volume of 25 mL with distilled water. Ca, Mg, Mn, Zn, Cu, and Fe were measured by flame atomic absorption, and Na, K by emission spectrometry (air-acetylene) using a novAA 350 apparatus (Analytik Jena, Jena, Germany).

### 2.4. Determination of Polyphenol Composition

Samples were prepared for HPLC analysis according to [20]. Briefly, 0.5 g honey were dissolved in 1 mL methanol using a vortex mixer and clarified by centrifugation at $12.000\times g$. Analysis of polyphenols was carried out by RP-HPLC using detection at 254 nm and an Ultrasphere ODS column (4.6 mm $\times$ 25 mm, 5 μm particle size) (Beckman-Coulter, Brea, CA, USA). Elution was at 1 mL/min using the following gradient of methanol in $H_2O$ (pH 3 with phosphoric acid): 0–70 min, gradient from 0 to 70% methanol; 70–75 min, 70% methanol; 75–80 min gradient from 70–100% methanol; 80–85 min, 100% methanol. Identifications were carried by LC/HRMS using an Agilent 1200 Rapid Resolution HPLC interfaced to a Bruker maXis mass spectrometer as described [21]. The detected compounds were identified by comparison to those of Chapman and Hall Dictionary of Natural Products database and authenticated using commercial standards.

### 2.5. Antimicrobial Activity

Antimicrobial activity was determined as the Minimum Inhibitory Concentration (MIC) and the Minimum Bactericidal Concentration (MBC). Three antibiotic-resistant strains of bacteria were used: *Escherichia coli* (466), a Gram-negative *Bacillus* resistant to cefuroxime, ceftriaxone, amoxicillin, ceftazidime, cefotaxime, cephalothin and ciprofloxacin; *Pseudomonas aeruginosa*, a Gram-negative *Bacillus* resistant to trimethoprim-sulphamethoxazole and amoxicillin/clavulanate; and *Staphylococcus aureus*, a Gram-positive cocci resistant to vancomycin. These bacteria were isolated and identified at the University Hospital Hassan II (Fez, Morocco). Bacteria were cultured at 37 °C in Mueller–Hinton Broth (MHB) liquid media and Mueller–Hinton Agar (MHA). Bacterial suspensions (inoculum) were obtained from sample colonies after 24-h cultures. These colonies were suspended in 0.9% NaCl, shaken for 15 s, and density was adjusted to 0.5 McFarland turbidity, which corresponds to an optical density of 0.08–0.13 at 625 nm. The final concentration was approximately 108 CFU (colony forming unit)/mL [22].

MIC was determined by micro-dilution tests in 96 well microplates according to the National Committee for Clinical Laboratory Standards (NCCLS) using honey (7.81 mg/mL to 250 mg/mL) and inoculum ($5 \times 10^5$ CFU/mL) in a final volume of 200 μL. After incubation for 20 h at 37 °C, 40 μL triphenyltetrazolium chloride (TTC) was added to each well. Bacterial growth is indicated by reduction of TTC to a red colored formazan. MIC is defined as the honey concentration that does not produce any red color after incubation for 2 h [22]. An aliquot from each well corresponding to a concentration equal or lower than MIC was sub-cultured on Mueller–Hinton agar (MHA) and incubated for 24 h at 37 °C in order to determine MBC, which is defined as the lowest honey concentration that does not allow bacterial growth [23].

### 2.6. Statistical Analysis

Means were compared by one-way ANOVA followed by post-hock Tukey test using IBM SPSS statistics version 25. Differences were considered statistically significant when $p < 0.05$. Independent student t-tests were used for two by two comparisons. Correlations between honey bioactivity and physicochemical parameters were determined using the Pearson correlation *r* value. These *r* values were grouped using the dendrogram function of MATLAB 2018a software and presented as a colored heat map.

Multivariate analysis by principal component analysis (PCA) was carried out using the MATLAB 2018a software. Only the most important parameters discriminating Zantaz honey samples were analyzed using PCA. Parameters were selected by plotting individual PCA for each group of parameters (standard physicochemical properties, sugars, minerals, and phenolic compounds). Only parameters participating in the discrimination of more than 500 of 1000 as a maximum (arbitrary unit), to one of the first three components were selected. Those selected subgroups of parameters were concatenated, normalized to avoid scale variability effect, and used to perform the PCA.

## 3. Results

### 3.1. Melissopalynology and Physicochemical Characterization

Pollen analysis was carried out in order to confirm the botanical origin of the Zantaz honey samples. The inclusion of honey of other floral origins was avoided by selecting samples in which pollen from *B. spinosum* represented at least 50% total pollen. Pollen from *Cytisus* sp. (Fabaceae) and *Populus* sp. (Salicaceae) were also found at low concentrations (Table 1).

**Table 1.** Average physicochemical characteristics of the Zantaz honey samples. Results are the average ± SD of ten samples.

| | | Means ± SD | Min | Max |
|---|---|---|---|---|
| Pollen analysis (%) | *Bupleurum spinosum* | 62 ± 10 | 51 | 80 |
| | *Cytisus* sp. | 12 ± 16 | 0 | 38 |
| | *Populus* sp. | 8 ± 5 | 1 | 19 |
| | Others | 18 ± 13 | 3 | 41 |
| General physicochemical properties | pH | 4.05 ± 0.21 | 3.78 | 4.38 |
| | Water content (%) | 19.88 ± 0.85 | 18.40 | 21.13 |
| | Free acidity (mEq/kg) | 18.41 ± 5.72 | 8.40 | 26.40 |
| | Lactonic acidity (mEq/kg) | 11.30 ± 1.36 | 8.00 | 13.00 |
| | Total acidity (mEq/kg) | 29.71 ± 6.66 | 18.40 | 38.40 |
| | Ash content (%) | 0.19 ± 0.05 | 0.13 | 0.32 |
| | Electrical conductivity (EC) (µs/cm) | 454.80 ± 85.50 | 351.66 | 652.33 |
| | Diastase activity (Shad number) | 20.21 ± 4.86 | 12.38 | 29.52 |
| | Melanoidins (a. u.) | 1.01 ± 0.28 | 0.40 | 1.45 |
| | Color (mm Pfund) | 62.71 ± 18.98 | 25.98 | 96.42 |
| Sugars (g/100 g) | Fructose | 38.50 ± 2.37 | 34.71 | 41.83 |
| | Glucose | 23.25 ± 1.96 | 19.61 | 26.89 |
| | Melibiose | 2.80 ± 0.92 | 1.44 | 4.31 |
| | Turanose | 1.79 ± 0.20 | 1.29 | 2.04 |
| | Maltose | 1.53 ± 0.47 | 0.95 | 2.45 |
| | Arabinose | 1.60 ± 0.72 | 0.33 | 2.44 |
| | Trehalose | 0.82 ± 0.29 | 0.50 | 1.50 |
| | Xylose | 0.30 ± 0.03 | 0.25 | 0.36 |
| | Sucrose | <0.2 | <0.2 | <0.2 |
| Minerals (mg/Kg) | K | 638.19 ± 206.29 | 429.15 | 1177.47 |
| | Ca | 154.78 ± 20.90 | 124.15 | 189.95 |
| | Na | 57.70 ± 15.46 | 38.39 | 90.90 |
| | Mg | 33.16 ± 9.55 | 24.25 | 52.80 |
| | Fe | 14.22 ± 3.36 | 7.61 | 18.23 |
| | Cu | 1.29 ± 0.36 | 0.89 | 1.95 |
| | Mn | 0.85 ± 0.18 | 0.51 | 1.12 |
| | Zn | 0.66 ± 0.42 | 0.26 | 1.72 |

The physicochemical characteristics of Zantaz honey are also shown in Table 1. None of the analyzed samples showed any abnormal values as compared to international quality standards [24,25]. Values for pH ranged from 3.78 to 4.38, which is important because acidic pH prevents the growth of microorganisms in honey [26]. Average water content was also below the maximum 20% allowed for honey [24,25]. Contents in water higher than this favor the growth of microorganisms responsible for fermentations that decrease the quality of honey. Free acidity was below 50 mEq/kg, and ash content and conductivity did not exceed the established limits of 0.6% and 800 µS/cm, respectively. Diastase activity provides an estimate of honey freshness; low values indicating inadequate storage or processing conditions. Diastase activity of all samples were well above the required 8 shade units [25]. Honey color and melanoidins are also important in the characterization of honey. Both parameters are estimated by measuring net absorbance of honey solutions and are linked to honey bioactivity. Average values in Zantaz honey samples were 1.01 and 62.71 mm Pfund for melanoidins and color, respectively.

Sugars represent up to 80 g/100 g, and the most abundant sugars in the Zantaz honey samples are fructose and glucose at 38.5 and 23.25 g/100 g, respectively. These results are consistent with those previously reported by others [27,28]. The concentration of sucrose, whose presence may indicate adulteration, was below 5 g/100 g as required by international standards except for honey from *Lavandula* that can reach up to 15 g sucrose/100 g. Two monosaccharides and four disaccharides representing not more than 3 g/100 g total composition were also found. Mineral content is an indicator of the geographical origin of honey and may also reveal potential environmental pollution if toxic metals are detected [29]. Potassium was the most abundant element in Zantaz honey at an average concentration of 638.19 mg/kg, followed by calcium with an average value of 154.78 mg/kg. The physicochemical characteristics of these Zantaz honey samples are in line with those previously reported [30].

The average phenolic composition of Zantaz honey samples is shown in Figure 2. The most abundant phenol was methyl syringate, which is consistent with previous work [15]. High contents of methyl syringate have also been reported in other honeys, including Sardinian *Asphodelus microcarpus* (Liliaceae) monofloral honey [31], and New Zealand Manuka and Kanuka honeys [32]. Epicatechin, syringic acid, and catechin were also abundant in Zantaz honey (Figure 2).

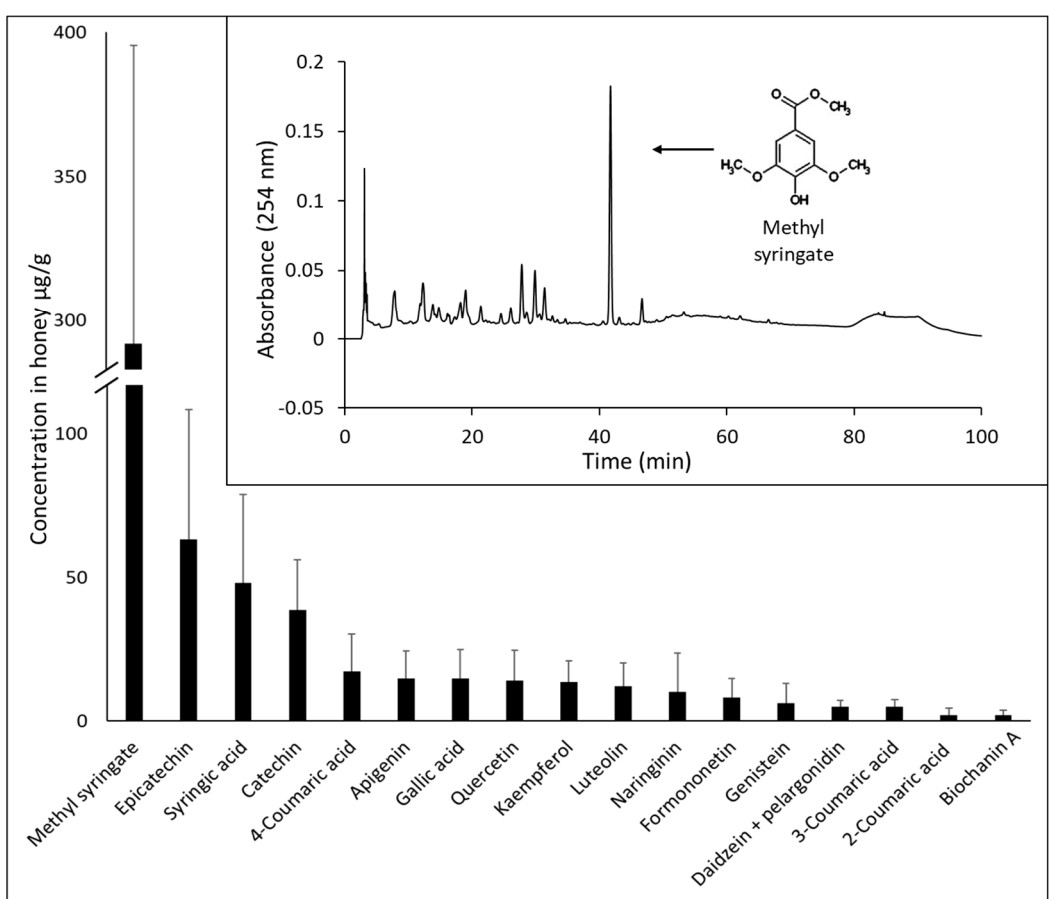

**Figure 2.** Polyphenol composition of Zantaz honey. Values are averages $\pm$ SD of 10 samples. Inset presents a typical RP-HPLC polyphenol profile and the structure of methyl syringate.

### 3.2. Antibacterial Activity

The three strains of bacteria that have been used to determine antibacterial activity behaved differently upon exposure to the honey samples (Table 2). *S. aureus* was the most sensitive to the treatments, with MIC and MBC values of 62.5 mg/mL for samples ZH2,

ZH7, ZH8, and ZH9. Sample ZH6 was the least effective with MIC and MBC values of 250 mg/mL. *P. aeruginosa* showed an intermediate sensitivity to exposure to the honey samples, and the threshold of 62.5 mg/mL MIC and MBC was reached only with honey samples ZH7 and ZH9. Sample ZH6 was also the least effective against *P. aeruginosa*. *E. coli* was the most resistant bacteria, and the threshold of 62.5 mg/mL was not reached by any of the Zantaz honey samples. This bacterium was resistant to most samples, with MIC and MBC values of 250 mg/mL for samples ZH4, ZH5, ZH6, and ZH10. Similar results were reported by others [12,33].

**Table 2.** Antibacterial activity of Zantaz honey against *E. coli*, *P. aeruginosa* and *S. aureus*.

| Zantaz Honey Samples | *E. coli* (466) | | *P. aeruginosa* | | *S. aureus* | |
|---|---|---|---|---|---|---|
| | MIC (mg/mL) | MBC (mg/mL) | MIC (mg/mL) | MBC (mg/mL) | MIC (mg/mL) | MBC (mg/mL) |
| ZH1 | 125 | 125 | 125 | 125 | 125 | 125 |
| ZH2 | 125 | 125 | 62.5 | 125 | 62.5 | 62.5 |
| ZH3 | 125 | 250 | 125 | 125 | 125 | 125 |
| ZH4 | 250 | 250 | 125 | 125 | 125 | 125 |
| ZH5 | 250 | 250 | 125 | 250 | 125 | 125 |
| ZH6 | 250 | 250 | 250 | 250 | 250 | 250 |
| ZH7 | 125 | 125 | 62.5 | 62.5 | 62.5 | 62.5 |
| ZH8 | 125 | 125 | 125 | 125 | 62.5 | 62.5 |
| ZH9 | 125 | 250 | 62.5 | 62.5 | 62.5 | 62.5 |
| ZH10 | 250 | 250 | 125 | 125 | 125 | 125 |
| Mean | 175 | 200 | 118.75 | 137.5 | 112.5 | 112.5 |
| SD | 61.2 | 61.2 | 51.9 | 61.2 | 54.5 | 54.5 |
| Min | 125 | 125 | 62.5 | 62.5 | 62.5 | 62.5 |
| Max | 250 | 250 | 250 | 250 | 250 | 250 |

## 4. Discussion

*Multivariate Analysis and Correlation between Zantaz Honey Composition and Antibacterial Activity*

The antibacterial activity of honey depends on its physicochemical characteristics and the susceptibility of the target bacteria [8]. Statistical tools such as correlation and multivariate analysis are used in order to establish relationships between physicochemical parameters and antibacterial activity. Pearson correlation was calculated for each physicochemical parameter and the antibacterial activity of the Zantaz honey samples included in this study. Values of r were transformed into a color scale in order to simplify the output, and correlations were clustered using a dendrogram presentation heat map model (Figure 3). The dendrogram in Figure 3a shows that all physicochemical parameters, except moisture, have a negative correlation with MIC at least for two of the bacterial strains, thus being potential contributors to the antibacterial activity of Zantaz honey.

Laallam et al. (2015) reported a positive correlation between water content and antibacterial activity in honeys from the Sahara region [13], despite the fact that an increase in water content results in dilution of antibacterial components, including sugars responsible for the antibacterial activity that is attributed to osmotic pressure in honey. The concept of concentrated and diluted nectars may explain this discrepancy. Thus, more concentrated nectars rich in antibacterial compounds such as those in the Sahara region would result in a positive correlation between water content of honey and antibacterial activity. In contrast, weak nectars with a higher content of water would have a diluting effect in antibacterial components, causing a negative correlation between water content and antibacterial activity, which appears to be the case in Zantaz honey.

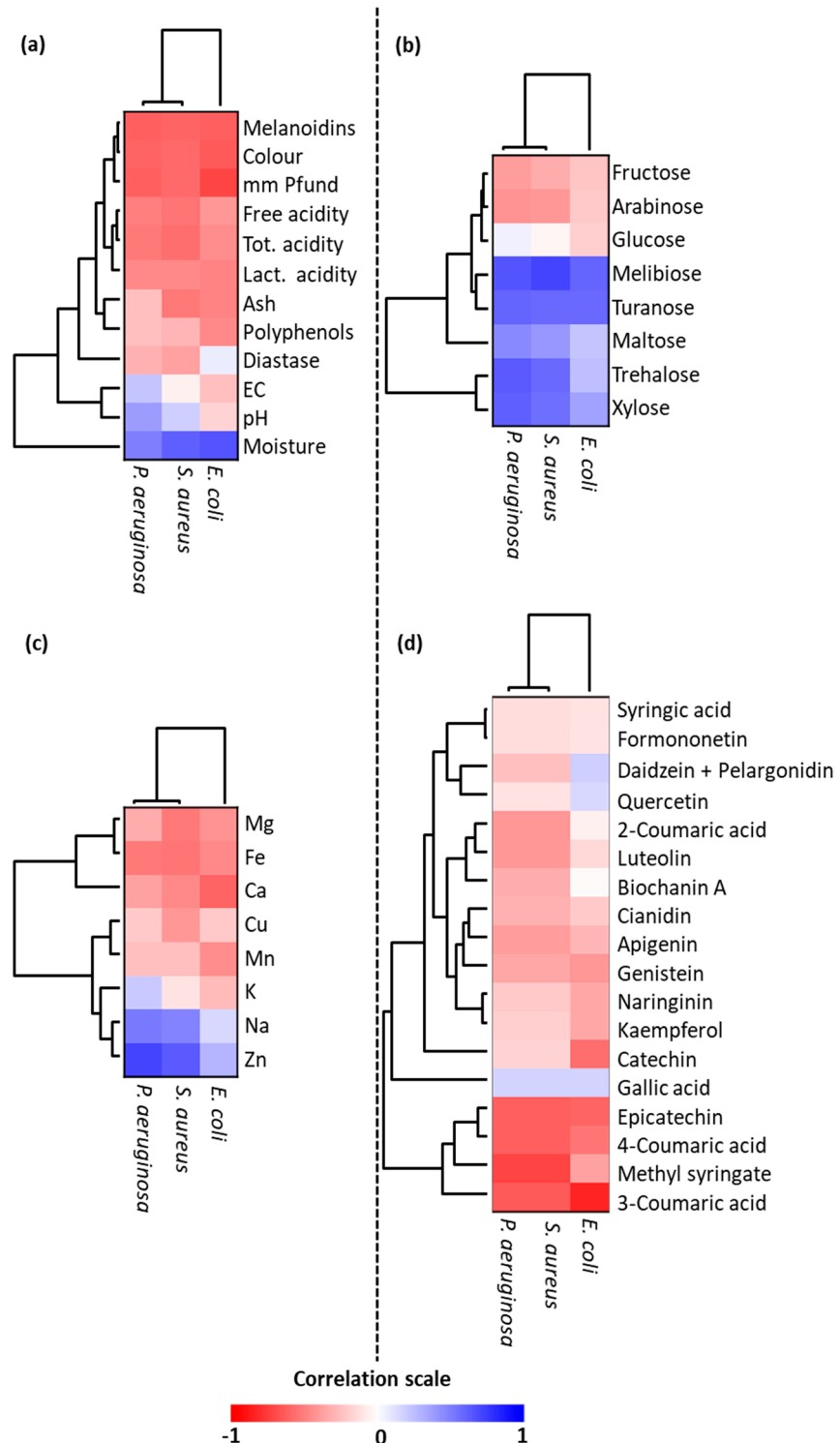

**Figure 3.** Heat map and dendrogram illustrating the correlation between physicochemical parameters and antimicrobial activity (MIC values) for *E. coli*, *P. aeruginosa* and *S. aureus*. (**a**) Physicochemical properties, (**b**) sugar composition, (**c**) mineral composition, (**d**) polyphenol composition.

Antibacterial activity had a positive correlation with the content of melanoidins and acidity, which is consistent with previous reports indicating the importance of melanoidins [19] and acidity [11,12] for antibacterial activity. A negative correlation between MIC and ash content was observed in *E. coli* and *S. aureus* and *P. aeroginosa*, suggesting a contribution of minerals to antibacterial activity. MIC values appeared to also correlate with content of

total polyphenols, although the correlation was only significant for *E. coli*. Previous works have reported that the antibacterial activity in honey is related to the presence of phenolics [34,35]. The color of honey reflects its richness in melanoidins and bioactive compounds such as polyphenols [36]. Hence, a positive correlation of color with antibacterial activity was also observed. The monosaccharides fructose, glucose, and arabinose are potential contributors to the antibacterial activity. Indeed, a negative correlation between MIC and the concentration of these sugars was observed. This may be at least partly related to the changes in the bacterial quorum sensing that bacteria have been reported to induce but also to sugars' osmotic pressure that affects bacterial growth, proliferation and formation of biofilms [37].

Glucose was only effective against *E. coli* (Figure 3b). The antibacterial activity of sugars appears to be due to osmotic dehydration caused by their very high concentration in honey [14,38]. However, glucose, which is the second most abundant sugar in Zantaz honey, showed little correlation with antibacterial activity. There is a negative correlation between the contents in fructose and glucose in Zantaz honey because these sugars may be coming from different floral sources (Figure 4B). A positive correlation with bacterial growth was found for several disaccharides and xylose, which is an aldopentose monosaccharide.

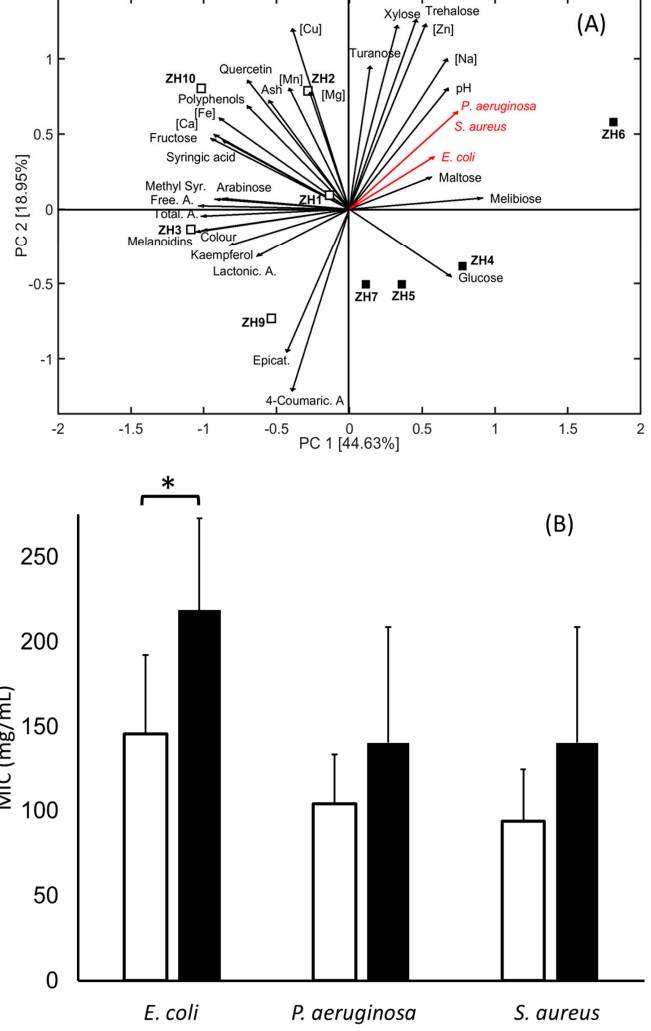

**Figure 4.** (**A**) PCA of Zantaz honey samples using the most discriminating parameters and MIC antibacterial activity (red arrows). (**B**) Average MIC of the groups with the lowest (full bars) and highest (open bars) antibacterial activity. * $p < 0.05$.

All minerals correlated with antibacterial activity except for Na and Zn (Figure 3c). Potassium, which is the most abundant mineral in Zantaz honey, showed the same behavior as EC, i.e., positive correlation with antibacterial activity against *E. coli* and *S. aureus*, and negative correlation against *P. aeruginosa*. Laallam et al. (2015) observed a negative correlation between electrical conductivity and antibacterial activity against *Bacillus subtilis*, *Clostridium perfringens*, *E. coli* and *S. aureus* in Algerian honeys, considering all strains as one single variable [13]. However, we have observed that this correlation is highly dependent on the strain of bacteria that is used.

Analysis of the correlation between antibacterial activity and phenolic composition revealed that polyphenols have antibacterial activity (Figure 3d). The highest positive correlation was observed in the cluster formed by epicatechin, 4-coumaric acid, methyl syringate, and 3-coumaric acid (Figure 3d). Methyl syringate was the most abundant polyphenol in Zantaz honey. Kirkpatrick, Nigam, and Owusu-Apenten (2017) determined using the disc diffusion technique that methyl syringate did not have antibacterial activity against *S. aureus*, *E. coli*, and *Bacillus subtilis* [39]. In our case, the correlation between methyl syringate content and antibacterial activity that has been found may be due to association of methyl syringate with other compounds, and/or to synergistic effects of methyl syringate and other components in Zantaz honey. In any case, our data shows that methyl syringate is a good marker for antibacterial activity in Zantaz honey.

The power of the physicochemical characteristics to predict antibacterial activity in Zantaz honey was analyzed by PCA. First and second principal components accounted for 63.58% of total variance (Figure 4A). Based on PC1, variables can be divided into two main groups. A first group correlated positively with PC1 and is not likely involved in antibacterial activity, because there is a positive correlation between parameters in this first group and MIC. These group includes glucose, melibiose, maltose, pH, sodium, zinc, and with lower contribution trehalose, xylose, and turanose. The second group of variables, which contributed to antibacterial activity, includes mainly melanoidins, free and total acidity, fructose, and methyl syringate. These results are in agreement with the correlation analysis showed in Figure 3. According to PCA, the Zantaz honey samples can be divided into two groups. Samples ZH4, ZH5, ZH6 and ZH7 (black squares) have the lowest antibacterial activities, and samples ZH1, ZH2, ZH3, ZH9 and ZH10 (white squares) have the highest antibacterial activities, considering their MIC values.

In order to test whether PCA discrimination may be used to predict antibacterial activity, average MIC were determined for both Zantaz honey samples groups and for each bacterial strain independently (Figure 4B). Results show that, based on physicochemical characteristics, Zantaz honey samples can be classified according to their antibacterial activity as indicated by the PCA. Hence, samples in group 1 (black bars), showed lower antibacterial activity (high MIC) against all bacterial strains in comparison with samples in group 2 (white bars).

## 5. Conclusions

In conclusion, the antibacterial activity of Zantaz honey has been investigated for the first time, and the correlation of this activity with physicochemical parameters has been determined. Our data show that this honey, which is rich in methyl syringate, has bactericidal properties against several bacterial strains that are resistant to antibiotics. Multivariate analysis indicated that melanoidins, acidity, phenolic compounds, hexoses, and divalent minerals are the physicochemical characteristics that best correlate with antibacterial activity. These results, together with previous reports describing the antioxidant and antiproliferative activities of Zantaz honey, may translate into a revalorization of this honey as a functional food with health-promoting properties. This revalorization would have very positive repercussions in the local rural communities from Morocco in which Zantaz honey is produced.

**Author Contributions:** Conceptualization, Y.E., M.G.M., H.I., B.L. and J.V.; methodology, Y.E., H.I., O.A., L.M.E., J.M. and M.A.; software, Y.E. and J.V.; writing—original draft preparation, Y.E., H.I. and J.V.; writing—review and editing, J.V., M.G.M., J.G.-C., O.A. and L.M.E.; funding acquisition, J.V. and O.A. All authors have read and agreed to the published version of the manuscript.

**Funding:** This research was funded by the C.S.I.C. Program for Scientific Cooperation for Development (Program I-COOP + 2018) and CEF-Forest Research Centre is a research unit funded by Fundação para a Ciência e a Tecnologia I.P. (FCT), Portugal: UIDB/00239/2020 and. UIDB/05183/2020.

**Institutional Review Board Statement:** Not applicable.

**Informed Consent Statement:** Not applicable.

**Data Availability Statement:** Not applicable.

**Acknowledgments:** Thanks are due to Fernando Reyes (Fundación MEDINA, Granada, Spain) for the identification of polyphenols in Zantaz honey. Thanks are due to Natália Roque from the GIS Laboratory of Polytechnique Institute of Castelo Branco for the map production.

**Conflicts of Interest:** The authors declare no conflict of interest. The funders had no role in the design of the study; in the collection, analyses, or interpretation of data; in the writing of the manuscript, or in the decision to publish the results.

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
