# Peer review of "Antibacterial Activity of Moroccan Zantaz Honey and the Influence of Its Physicochemical Parameters Using Chemometric Tools"

_applsci, doi:10.3390/app11104675_

Round 1

Reviewer 1 Report

The manuscript is pretty interesting, with promising significance for recognition of Moroccan Zantaz honey, as a valuable source of health-promoting bioactive compounds.

The manuscript is well written, particularly interesting Introduction section.

According to my opinion, the article should be accepted for publication.

In Table 2 authors show MIC for 10 different honey samples, which differ in MIC/MBC. On the other side, the composition parameters are given as mean values (st dev) of 10 samples. It should be interesting to see the composition of samples that differ in MIC/MBC.

In line 171 r – italic; Table 1 – unit mEq/kg instead of mEq/Kg

Author Response

The manuscript is pretty interesting, with promising significance for recognition of Moroccan Zantaz honey, as a valuable source of health-promoting bioactive compounds.

The manuscript is well written, particularly interesting Introduction section.

According to my opinion, the article should be accepted for publication.

In Table 2 authors show MIC for 10 different honey samples, which differ in MIC/MBC. On the other side, the composition parameters are given as mean values (st dev) of 10 samples. It should be interesting to see the composition of samples that differ in MIC/MBC.

à The authors agree with the reviewer’s comment. However, our strategy of presenting the physicochemical data as Means±sd was because Zantaz characterization was published several times before and the data reported here did not show any worth-mentioning difference with the previous ones.

In line 171 r – italic; Table 1 – unit mEq/kg instead of mEq/Kg

The comment was considered and corrected in table 1.

Reviewer 2 Report

Antibacterial activity of Moroccan Zantaz honey and the influence of its physicochemical parameters using chemometric tools

The overall goal of this manuscript was to determine the antibacterial properties of Zantaz honey produced from the Moroccan Atlas Mountains against E. coli, P. aeruginosa, and S. aureus. Results of the study show that MBC and MIC of S.aureus were the lowest indicating that this species is most sensitive to Zantaz honey. Certain parameters associated with antibacterial activity include color, acidity, fructose, melanoidins content, epicatechin, methyl syringate, 4-coumaric acid, and 3-coumaric acid.

The manuscript requires some English language copyediting. The introduction was well written and discussed all aspects needed to know about the goal of the project.

It will be ideal to show the geographical sources of various honey samples. For example, it is suggested that a map showing the various locations of honeys be provided. How does one ensure that such samples are represented from diverse sources? Further, what are the threshold requirements for honey samples to be considered effective against the aforementioned bacteria. The dendograms provided (Figure 2) should have a control group. How do you ensure that systematic biases related to the representation in the dendograms? The explanation on the rationale behind melissopalynology was not thoroughly explained. Lastly, how do you ensure that samples coming from the source are homogenously represented?

Overall, the manuscript is well written and captured the important aspects for the journal, Applied Sciences.

Author Response

Antibacterial activity of Moroccan Zantaz honey and the influence of its physicochemical parameters using chemometric tools

The overall goal of this manuscript was to determine the antibacterial properties of Zantaz honey produced from the Moroccan Atlas Mountains against E. coli, P. aeruginosa, and S. aureus. Results of the study show that MBC and MIC of S.aureus were the lowest indicating that this species is most sensitive to Zantaz honey. Certain parameters associated with antibacterial activity include color, acidity, fructose, melanoidins content, epicatechin, methyl syringate, 4-coumaric acid, and 3-coumaric acid.

The manuscript requires some English language copyediting. The introduction was well written and discussed all aspects needed to know about the goal of the project.

It will be ideal to show the geographical sources of various honey samples. For example, it is suggested that a map showing the various locations of honeys be provided. How does one ensure that such samples are represented from diverse sources?

à The main aim of the present manuscript was to elucidate the influence of the physicochemical properties on the antibacterial activity of Zantaz honey regardless of their geographical origin. Hence, the used multivariate analysis focused on that. To minimize the influence of geographical region, and the possible inclusion of other honey types, only samples with high Bupleurum spinosum (Zantaz) pollen were considered. However, information about the geographical regions was included in the revised version. Please check the sampling subsection of material and methods.

Further, what are the threshold requirements for honey samples to be considered effective against the aforementioned bacteria.

à Generally, honey is applied for topic uses. an use that is linked to its antimicrobial activity. Each region usually uses the available honey. Hence, the aim of the present work was to investigate for the first time the antibacterial activity of Zantaz honey and try to determine the parameters that most influence this activity. This step is crucial to select the best sample from a group of characterized samples. In addition, dealing with several variable, all with possible influence on the antibacterial activity of honey make the establishment of such threshold a complicated task. Hence, the conclusion was limited to determine the parameters that needs to be maximized every time a selection process is required. 

The dendograms provided (Figure 3) should have a control group. How do you ensure that systematic biases related to the representation in the dendograms?

à We did not understand what the reviewer meant by “control group”. If the bias mentioned by the reviewer was regarding the possible inclusion of other botanical sources, we highlight that the samples selected for the present work were those that presented high percentage of Bupleurum spinosum pollen. Hence possible bias in the correlation coefficient of other botanical sources was reduced. 

The explanation on the rationale behind melissopalynology was not thoroughly explained. Lastly, how do you ensure that samples coming from the source are homogenously represented?

àThe mellissopalynology in the present work was accomplished with the aim of confirming the botanical source of the analysed samples. Only samples with high percentage of Bupleurum spinosum pollen were selected. Previous work of the same research group has established 45% of Bupleurum spinosum pollen, as monoflorality threshold. For the present work the average was above this (62 ± 10 %) to avoid the inclusion of honey from other sources, and make the conclusions as limited as possible to Zantaz honey.

Overall, the manuscript is well written and captured the important aspects for the journal, Applied Sciences.

We hope that we have adequately addressed all the reviewers’ remarks and questions, and that the manuscript is now suitable for publication.